# An Explainable 3D Convolutional Neural Network with Reliable Feature Selection and Hybrid 3D Image Block Ranking

## Abstract

An explainable 3D imaging challenge is effectively interpreting the rational relationship among top-ranked 3D image blocks, relevant features selected from extracted 3D feature maps, and decisions of a 3D convolutional neural network (CNN). We propose an explainable 3D CNN that integrates a robust feature selection (FS) method with a new hybrid block ranking algorithm to uncover the spatial relationship among 3D image blocks, selected features, and clinical diagnosis. The 3D image block ranking pipeline begins with a multi-FS procedure that removes irrelevant features that are out of an object and produces diverse selected feature sets. Then the selected feature sets are used to construct complementary 3D distribution, ranking, and average-ranking feature matrices that quantify importance levels for both relevant 3D image blocks and decisions. Finally, the novel hybrid 3D image block ranking algorithm leverages the complementary 3D feature matrices to reliably generate a block ranking map (BRM) with 3D block ranking numbers. Simulation results using 982 3D ADNI data for Alzheimer's disease (AD) diagnosis (3-class classification) indicate that the top-ranked 3D image blocks contain brain areas associated with AD diagnosis for two different simulations using overlapping neighboring and non-overlapping neighboring 3D image blocks. A medical doctor may conveniently use the BRM's top-ranked 3D image blocks of a patient's 3D image visualized by axial 2D patches, coronal 2D patches, and sagittal 2D patches to efficiently make an explainable and correct medical diagnosis. Importantly, the new 3D CNN with FS is better than the traditional 3D CNN without FS in terms of test accuracy, F1-score, AUC, and model size for both AD diagnosis and autism diagnosis (binary classification). Thus, the new 3D CNN with the reliable FS and the hybrid 3D image block ranking is more effective than the traditional 3D CNN for explainable, accurate, and memory-efficient 3D image classification applications.

## 1 Introduction

The 2D Convolutional Neural Network (CNN) Krizhevsky et al. (2012); LeCun et al. (2015); He et al. (2016) are applied in different explainable 2D image classification applications Zhou et al. (2016); Selvaraju et al. (2017); Zhang et al. (2018); Schöttl (2022); Wang et al. (2022), such as medical image classification. In recent years, 3D CNNs have been used in explainable image classification applications Morita et al. (2024); Wei et al. (2023); Zhang et al. (2020); Yang et al. (2018). For example, the 3D residual self-attention deep neural network is developed to capture local, global, and spatial information of MR images, improve diagnostic performance, and provide an explainable mechanism for identifying important brain regions Zhang et al. (2020).

However, there are new problems in explainable 3D deep learning. The first significant challenge is how to discover and interpret the rational relationship among 3D image blocks, features selected from extracted 3D feature maps, and decisions of a 3D deep learning model. Convolutional layers generate a feature in an extracted 3D feature map by using voxels in a relevant $R_h \times R_w \times R_d$ 3D image block where $R_h$, $R_w$, and $R_d$ are receptive fields Araujo et al. (2019); Luo et al. (2016). Since the features used by a classifier to make a final decision have useful informative properties, such as feature rankings, the informative feature properties can be used to rank the relevant $R_h \times R_w \times R_d$

3D image blocks. It is useful to investigate the relationship between ranked 3D image blocks and the decisions of a 3D deep learning model for explainable 3D imaging applications.

The second significant challenge is how to build an effective 3D CNN for explainable 3D image classification applications. For example, although the six most important brain regions associated with AD diagnosis (Hippocampus, Entorhinal Cortex, Cerebral Cortex (Temporal, Parietal, and Frontal Lobes), Temporal Lobe, Parietal Lobe, and Frontal Lobe) Braak & Braak (1991); Gómez-Isla et al. (1996); Jack et al. (1997); Jacobs et al. (2012); Minoshima et al. (1997) are discovered, it is tedious and difficult for a doctor to view such large brain regions to efficiently make a correct diagnosis along with a reasoning behind the decision. A technical challenge is how to identify important 3D blocks that contain small brain areas associated with the brain disease diagnosis in the large 3D brain regions to allow a doctor to efficiently and effectively make a correct and explainable diagnosis. For instance, a $256 \times 256 \times 256$ brain image contains $32,768$ non-overlapping neighboring $8 \times 8 \times 8$ blocks, and it would be very difficult for doctors to identify top-ranked blocks from the $32,768$ blocks for a correct and explainable diagnosis. Thus, it is necessary to effectively identify small top-ranked 3D medical image blocks that are critical for decisions and then understand the relationship among the top-ranked 3D image blocks, top-ranked features, and medical diagnosis.

To increase the explainability of a 3D CNN, we develop a new 3D CNN with a new "Feature Filter (FF) Layer". Since the top-ranked features selected by a FS method are associated with both 3D input image blocks and decisions, we create a novel hybrid 3D image block ranking algorithm using newly defined 3D feature matrices with informative top-ranked features' properties. Finally, a new "Block Ranking Map" (BRM), a matrix containing 3D image blocks' ranking numbers associated with importance levels for explainable 3D image classification, is generated. The BRM is convenient for a user to understand the relationship among top-ranked 3D image blocks, relevant features and a final decision, such as medical diagnosis.

## 2 THE RELATIONSHIP AMONG 3D IMAGE BLOCKS, FEATURES IN FEATURE MAPS AND DECISIONS OF A 3D CNN

The last maxpooling layer of a 3D CNN generates $M$ $H \times W \times D$ feature maps $F^l$ from a $\bar{H} \times \bar{W} \times \bar{D}$ 3D input image having features $f_{ijk}^l$ for $i = 0, 1, \ldots, H - 1$, $j = 0, 1, \ldots, W - 1$, and $k = 0, 1, \ldots, D - 1$, and $l = 0, 1, \ldots, M - 1$. A feature $f_{ijk}^l$ in a feature map is associated with voxels in a relevant $R_h \times R_w \times R_d$ 3D image block ($B_{ijk}$).

A 3D CNN has $L$ convolutional layers, where layer $l$ ($l = 1$ to $L$) has a 3D kernel size $k^l = (k_H^l, k_W^l, k_D^l)$ (typically odd integers for centered kernels) and stride $s^l = (s_H^l, s_W^l, s_D^l)$, with $s^l \geq 1$. Assuming zero padding for computing receptive fields (as padding does not affect the receptive field size), based on the works Araujo et al. (2019); Luo et al. (2016), the receptive fields of a 3D image block are computed as: $R_h = 1 + \sum_{l=1}^{L}(k_H^l - 1)\prod_{i=1}^{l-1} s_H^i$, $R_w = 1 + \sum_{l=1}^{L}(k_W^l - 1)\prod_{i=1}^{l-1} s_W^i$, and $R_d = 1 + \sum_{l=1}^{L}(k_D^l - 1)\prod_{i=1}^{l-1} s_D^i$. Here, $\prod_{i=1}^{l-1} s_H^i$, $\prod_{i=1}^{l-1} s_W^i$, and $\prod_{i=1}^{l-1} s_D^i$ are the cumulative strides up to layer $l - 1$ (with the empty product for $l = 1$ defined as 1). For dilated convolutions with dilation rate $\alpha^l$, replace $k^l$ with the effective kernel size $(k^l - 1)(\alpha^l - 1) + 1$. For pooling layers (e.g., max-pooling with window $p^l$ and stride $q^l$), treat them as convolutions with $k^l = p^l$, $s^l = q^l$. Different 3D CNN models may generate two different $R_h \times R_w \times R_d$ 3D image blocks that are overlapping neighboring 3D image blocks with shared voxels, and non-overlapping neighboring 3D image blocks without shared voxels.

A feature in an extracted feature map is generated by using voxels in a relevant $R_h \times R_w \times R_d$ 3D image block through multiple convolutional layers. Thus, the feature is a non-linear function of the voxels in a relevant $R_h \times R_w \times R_d$ 3D image block. An extracted feature map has $P$ flatten features that are associated with the relevant $P$ input 3D image blocks, and the $P$ flatten features are used by a classifier as inputs to make decisions. Thus, there is an important mathematical relationship among input 3D image blocks, extracted features, and final decisions.

Since all features in the $M$ feature maps are used by a traditional 3D CNN, each 3D image block is associated with $M$ features. Therefore, the 3D image blocks cannot be ranked based on the same number of associated features. To solve the problem of the traditional 3D CNN, we develop a new 3D CNN with the FF layer that generates a small number of top-ranked features that have been

selected by an offline FS method. A 3D image block associated with $M_1$ features is more important than another 3D image block associated with $M_2$ features for $M_1 > M_2$, $0 < M_1 \leq M$, and $0 \leq M_2 < M$. A 3D image block associated with 0 features (i.e., all $M$ associated features are eliminated by the FS method) is not useful for a decision of a 3D CNN with the FF layer. Thus, 3D image blocks can be ranked based on the number of associated top-ranked features. In addition, other feature properties, such as feature ranking scores, can be used to rank blocks. For example, if Block A with an average feature ranking score of 3.4 and the best feature ranking score of 2 is more important than Block B with an average feature ranking score of 6.7 and the best feature ranking score of 4. To reliably rank 3D image blocks, we define different informative 3D feature structures in the following section.

## 3    INFORMATIVE 3D FEATURE STRUCTURES

### 3.1    3D FEATURE SELECTION MAP

The $M$ feature maps are converted to $m$ flattened features for $m = n \times H \times W \times D$. The $m$ features have $m$ relevant feature index numbers $(0, 1, \ldots, m-1)$. A FS method selects the top $n$ features from the $m$ features. The $n$ selected features have $n$ feature index numbers $I_p$ for $I_p \in 0, 1, \ldots, m - 1$ for $p = 0, 1, \ldots, n-1$. A top feature with $I_p$ is associated with a feature map $F^{q_p}$ where $q_p = I_p \bmod n$ for $p = 0, 1, \ldots, n - 1$. Let $\bar{Q} = \{q_0, q_1, \ldots, q_{n-1}\}$. After eliminating duplicated elements in $\bar{Q}$, we get $Q$ with distinct elements for $Q \subseteq \bar{Q}$.

**Definition 1**: Let the 3D feature selection map $V^l$ be a 3D matrix that has elements $v_{ijk}^l$ for $i = 0, 1, \ldots, H - 1$, $j = 0, 1, \ldots, W - 1$, $k = 0, 1, \ldots, D - 1$, and $l = 0, 1, \ldots, M - 1$. If $f_{ijk}^l$ in a feature map $F^l$ is a feature selected by a FS method, then $v_{ijk}^l = f_{ijk}^l$, otherwise $v_{ijk}^l = 0$.

### 3.2    3D FEATURE MATRICES

To investigate the relationship among 3D image blocks, top features, feature selection maps, and final decisions, we define five new 3D feature structures as follows.

**Definition 2**: Let the "3D feature binary matrix" $B^l$ have binary numbers $b_{ijk}^l$ for $i = 0, 1, \ldots, H - 1$, $j = 0, 1, \ldots, W - 1$, $k = 0, 1, \ldots, D - 1$, and $l = 0, 1, \ldots, M - 1$. If $f_{ijk}^l$ is a selected feature, then $b_{ijk}^l = 1$, otherwise $b_{ijk}^l = 0$.

**Definition 3**: Let the "3D feature accumulation matrix" $A$ have elements called "3D feature accumulators" $a_{ijk}$ for $i = 0, 1, \ldots, H - 1$, $j = 0, 1, \ldots, W - 1$, and $k = 0, 1, \ldots, D - 1$, where $a_{ijk} = \sum_{l=0}^{M-1} b_{ijk}^l$ where $b_{ijk}^l$ is an element of the feature binary matrix $B^l$, and $M$ is the number of feature maps.

**Definition 4**: Let the "3D feature distribution matrix" $C$ have elements $c_{ijk}$ for $i = 0, 1, \ldots, H - 1$, $j = 0, 1, \ldots, W - 1$, and $k = 0, 1, \ldots, D - 1$, where $c_{ijk} = a_{ijk}/n$ where $a_{ijk}$ are feature accumulators of the feature accumulation matrix $A$, and $n$ is the number of selected features.

The features $v_{ijk}^q$ in the 3D feature selection map $V^q$ are ranked by a feature ranking method, such as the RFE Guyon et al. (2002); RFE (2025), then each feature has its ranking number $\hat{r}_{ijk}^q$ for $i = 0, 1, \ldots, H - 1$, $j = 0, 1, \ldots, W - 1$, $k = 0, 1, \ldots, D - 1$, and $q \in Q$, where the lower a ranking number, the higher a feature ranking. $\hat{r}_{ijk}^q$ are sorted to generate new ranking numbers $r_{ijk}^s$ in an increasing order for $s = 0, 1, \ldots, a_{ijk} - 1$.

**Definition 5**: Let the "3D feature ranking matrix" $R^l$ have positive integer ranking numbers $r_{ijk}^l$ for top-ranked features where $r_{ijk}^l \leq r_{ijk}^{l+1}$ for $l = 0, 1, \ldots, a_{ijk} - 1$, $i = 0, 1, \ldots, H - 1$, $j = 0, 1, \ldots, W - 1$, and $k = 0, 1, \ldots, D - 1$, where $a_{ijk}$ are 3D feature accumulators of the 3D feature accumulation matrix $A$. The smaller a positive integer ranking number, the higher the ranking of the top feature. Elements other than positive integer ranking numbers of $R^l$ are 0.

**Definition 6**: Let the "3D average feature ranking matrix" $\bar{R}$ have average feature ranking values $\bar{r}_{ijk}$ where $\bar{r}_{ijk} = (\sum_{l=0}^{a_{ijk}-1} r_{ijk}^l)/a_{ijk}$ for $i = 0, 1, \ldots, H - 1$, $j = 0, 1, \ldots, W - 1$, and $k = 0, 1, \ldots, D - 1$, where $a_{ijk}$ are 3D feature accumulators of the 3D feature accumulation matrix $A$.

## 4 The Hybrid 3D Image Block Ranking Algorithm

**Definition 7**: The "block ranking map" (BRM) is a $\bar{H} \times \bar{W} \times \bar{D}$ 3D matrix having $P$ blocks with positive block ranking numbers $\phi_{ijk}$ for $i = 0, 1, \ldots, H-1$, $j = 0, 1, \ldots, W-1$, $k = 0, 1, \ldots, D-1$, and $P = H \times W \times D$. The smaller $\phi_{ijk}$ is, the more important a block at $(i, j, k)$ is associated with the decision.

The 3D CNN with FS and hybrid 3D image block ranking, as shown in Fig. 3 in Appendix A, consists of eight components. The eight components include (1) 3D convolutional layers for extracting $M$ $H \times W \times D$ feature maps from a $\bar{H} \times \bar{W} \times \bar{D}$ 3D input image with $P$ $H \times W \times D$ blocks for $P = H \times W \times D$, (2) the flatten layer for converting $m$ flattened features from the $M$ $H \times W \times D$ feature maps for $m = n \times H \times W \times D$, (3) the FF layer for generating $n$ top-ranked features from the $m$ flattened features by using top-ranked feature index numbers (note: the top-ranked feature index numbers have been identified by an offline FS method), (4) the feature analyzer for generating 3D feature distribution matrices, 3D feature ranking matrices and 3D average feature ranking matrices, (5) a 3D image block ranking method for generating a BRM, (6) an image software tool, such as the "ebrains" software tool Atlas (2025) that can be used to get useful information from a 3D brain image, (7) diverse information from experts and literature is used to generate hybrid information to explain the relationship between blocks and decisions, and (8) the hybrid explainable information generator than can produce explainable decision-making information, such as relations among top-ranked blocks, relevant feature maps, relevant features, relevant brain areas, and a brain disease.

Since different FS methods generate different feature sets, it is necessary to develop a new multi-FS algorithm (Algorithm 1) to reliably discover top-ranked features.

---

**Algorithm 1** The Multi-FS Algorithm

---

**Input:** Training Data with $N$ features.
**Output:** Different sets of $K$ ranked selected features.
  1: Step 1: Eliminate features that are not associated with a rational image region defined by an expert to generate a pool of $U$ selected features for $U < N$.
  2: Step 2: Use different FS methods to generate different sets with $K$ features selected from the pool for $K < U$, respectively.
  3: Step 3: Apply a feature ranking method to rank $K$ selected features in each selected feature set.

---

A 3D image block associated with more top-ranked features, the best feature with a higher ranking score, and a higher average feature ranking score is more important for decision-making than a block with less top-ranked features, a lower-ranking best feature, and a lower average feature ranking score. Thus, a 3D feature distribution matrix, a 3D feature ranking matrix, and a 3D average feature ranking matrix can be used to rank blocks. A user, such as a medical doctor, can analyze the relationship among the top-ranked blocks, top-ranked features, relevant feature maps, and a final decision.

To reliably rank 3D image blocks, we develop a new hybrid 3D image block ranking algorithm (Algorithm 2) using different BRMs under different conditions.

## 5 Simulations Using Non-overlapping Neighboring 3D Blocks

The ADNI dataset with 982 brain images with different sizes ADNI (2025); Amin (2025), such as $192 \times 192 \times 160$ and $256 \times 256 \times 170$, is used for three-class 3D image classification performance analysis. The 982 3D brain images include 284, 477, and 221 3D brain images for the cognitively normal (CN) class, the mild cognitive impairment (MCI) class, and the AD class, respectively. The 982 3D images are resized to 982 $64 \times 64 \times 64$ images. The 982 resized 3D images are split into 785 training images and 197 testing images. A 3D CNN with two Conv3d layers (3×3×3, stride 1, padding 1) (each Conv3d layer followed by BatchNorm3d, ReLU, and a 2×2×2 max-pool) is trained by using the 785 training images. The second Conv3d layer generates 8 $16 \times 16 \times 16$ feature maps that are converted to $32,768$ flatten features. 785 new training data with the $32,768$ features are used for FS. The 3D CNN model generates features in feature maps that are associated with non-overlapping neighboring $4 \times 4 \times 4$ 3D image blocks.

---

**Algorithm 2** The hybrid 3D image block ranking algorithm

---

**Input:** New training data with the $m$ flattened features from the $M$ $H \times W \times D$ feature maps.
**Output:** The BRM with block ranking numbers.
 1: Step 1: Use the multi-FS algorithm to generate $\check{N}$ feature groups of $K_\alpha$-feature sets, and get $\bar{N}$ different feature sets $\check{F}_\beta^\alpha$ for $\alpha = 1, 2, \ldots, \tilde{N}$ where $\tilde{N}$ is the number of different feature sets, $\beta = 1, 2, \ldots, \check{N}$, and $\bar{N} = \check{N}\tilde{N}$.
 2: Step 2: Use or $\check{K}$ different feature ranking methods for $\check{K} \geq 2$ (use a feature ranking method for $\check{K} = 1$) to rank features in the $\bar{N}$ different feature sets to generate $\bar{K}$ different feature ranking sets for $\bar{K} = \check{K}\bar{N}$.
 3: Step 3: Generate $c_{ijk}^{\alpha\beta}$ in $\bar{N}$ 3D feature distribution matrices based on $\bar{N}$ different feature sets, calculate $r_{ijk}^{\alpha\beta\kappa}$ in $\bar{K}$ 3D feature ranking matrices based on $\bar{K}$ different feature ranking sets, and calculate $\bar{r}_{ijk}^{\alpha\beta\kappa})$ in $\bar{K}$ 3D average feature ranking matrices based on $\bar{K}$ 3D feature ranking matrices.
 4: Step 4: Calculate block ranking scores $\theta_{ijk}^{\alpha\beta\kappa}$ of a block at $(i, j, k)$ where the increasing function $\theta_{ijk}^{\alpha\beta\kappa} = f(c_{ijk}^{\alpha\beta}, r_{ijk}^{\alpha\beta\kappa}, \bar{r}_{ijk}^{\alpha\beta\kappa})$ for $\kappa = 0, 1, \ldots, \check{K}-1$, $i = 0, 1, \ldots, H-1$, $j = 0, 1, \ldots, W-1$, and $k = 0, 1, \ldots, D-1$.
 5: Step 5: Generate block ranking numbers $\lambda_{ijk}^{\alpha\beta\kappa}$ based on the block ranking scores $\theta_{ijk}^{\alpha\beta\kappa}$ .
 6: Step 6: Generate $BRM^{\alpha\beta\kappa}$ using the $\lambda_{ijk}^{\alpha\beta\kappa}$.
 7: Step 7: Generate the $BRM$ with average block ranking numbers $\bar{\lambda}_{ijk}$ where $\bar{\lambda}_{ijk} = \frac{1}{\bar{N}\check{N}\bar{K}} \sum_{\alpha=1}^{\tilde{N}} \sum_{\beta=1}^{\check{N}} \sum_{\kappa=1}^{\bar{K}} \lambda_{ijk}^{\alpha\beta\kappa}$ for $i = 0, 1, \ldots, H-1$, $j = 0, 1, \ldots, W-1$, and $k = 0, 1, \ldots, D-1$.
 8: Step 8: Generate block ranking numbers $\phi_{ijk}$ based on $\bar{\lambda}_{ijk}$.
 9: Step 9: Generate the final BRM using the $\phi_{ijk}$.

---

## 5.1 Detailed Steps of Algorithm 1

The Multi-FS algorithm with the three steps is used to select the top features from the $32,768$ features that have $32,768$ feature index numbers $F_p$ for $p = 0, 1, \ldots, 32,767$. The resized $64 \times 64 \times 64$ image contains $4,096$ non-overlapping neighboring $4 \times 4 \times 4$ blocks. A block has 3D indices $i$, $j$, and $k$ for $i = 0, 1, \ldots, 15$ along the top-to-bottom direction of a brain, $j = 0, 1, \ldots, 15$ along the front-to-back direction of a brain, and $k = 0, 1, \ldots, 15$ along the left-to-right direction of a brain. The $64 \times 64 \times 64$ image is partitioned to 16 axial brain image slices $S_{axial}^i$, 16 coronal brain image slices $S_{coronal}^j$, and 16 sagittal brain image slices $S_{sagittal}^k$ for $i = 0, 1, \ldots, 15$, $j = 0, 1, \ldots, 15$, and $k = 0, 1, \ldots, 15$. Each brain image slice has four brain images.

The first step of the multi-FS algorithm eliminates features that are not associated with the 3D brain image region to generate a pool of selected features. Firstly, block's 3D indices of the $32,768$ features are calculated: $i_p = \left\lfloor \frac{F_p \bmod 4,096}{256} \right\rfloor$, $j_p = \left\lfloor \frac{F_p \bmod 64}{4} \right\rfloor$, and $k_p = F_p \bmod 4$ for $p = 0, 1, \ldots, 32,767$. Secondly, if the block's 3D indices are within axial, coronal and sagittal index ranges of the 3D brain images, the relevant feature is added into a feature pool.

To generate the rational axial index ranges, the rational coronal index ranges, and the rational sagittal index ranges of the 3D brain images, we generated average axial brain images $\bar{S}_{axial}^i$ by averaging training brain images in $S_{axial}^i$ for $i = 0, 1, \ldots, 15$, average coronal brain images $\bar{S}_{coronal}^j$ by averaging training brain images in $S_{coronal}^j$ for $j = 0, 1, \ldots, 15$, and average axial brain images $\bar{S}_{sagittal}^k$ by averaging training brain images in $S_{sagittal}^k$ for $k = 0, 1, \ldots, 15$. For example, $\bar{S}_{axial}^1$, $\bar{S}_{axial}^3$, $\bar{S}_{axial}^5$, and $\bar{S}_{axial}^7$ are shown in Fig. 1(a), Fig. 1(b), Fig. 1(c), and Fig. 1(d), respectively.

Since features associated with black patches not located within the average brain images shown in Fig. 1 are not useful for explainable decision-making, they must be eliminated by feature selection rules. For example, a feature selection rule for $\bar{S}_{axial}^{i_p}$ for $i_p = 1$, as shown in Fig. 1(a), is: "If $(j_p = 5$ and $6 \leq k_p \leq 9)$ or $(j_p = 6$ and $5 \leq k_p \leq 10)$ or $(j_p = 7$ and $4 \leq k_p \leq 11)$ or $(j_p = 8$ and $4 \leq k_p \leq 11)$ or $(j_p = 9$ and $4 \leq k_p \leq 11)$ or $(j_p = 10$ and $4 \leq k_p \leq 11)$ or $(j_p = 11$ and $5 \leq k_p \leq 10)$ or $(j_p = 12$ and $6 \leq k_p \leq 9)$, Then the feature is added into the feature pool."

A complete feature selection rule is: "If a feature with a feature index number $F_p$ is associated with a patch at $(i_p, j_p, k_p)$ within the average axial brain $\bar{S}^{i_p}_{axial}$, the average axial brain $\bar{S}^{j_p}_{coronal}$, and the average axial brain $\bar{S}^{k_p}_{sagittal}$ for $p = 0, 1, \ldots, 32, 767$, Then the feature is added into the feature pool." It is used to eliminate $22,864$ features ($69.8\%$ of all features) from the $32,768$ features, and generate the pool of $9,904$ remaining features for further FS. The $22,864$ eliminated features are associated with the black regions that are not within the average brain images, as shown in Fig. 1.

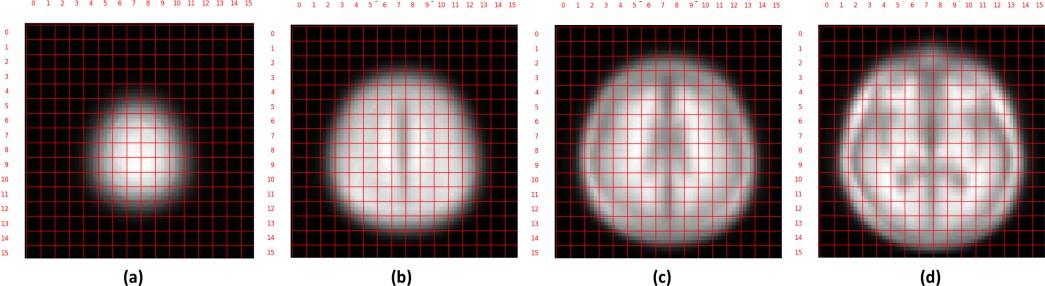

(a)  (b)  (c)  (d)

Figure 1: Four average axial brain images including (a) $\bar{S}^1_{axial}$, (b) $\bar{S}^3_{axial}$, (c) $\bar{S}^5_{axial}$, and (d) $\bar{S}^7_{axial}$ where the row number is $j_p$, and the column number is $k_p$.

The second step of the multi-FS algorithm uses five basic FS methods (Chi2 scikit-learn developers (2025a), mutual_info_classif scikit-learn developers (2025d), f_regression scikit-learn developers (2025c), f_classif scikit-learn developers (2025b), and the RFE method) in 12 different ways to select 4 different feature groups $FG_u$ for $u = 0, 1, \ldots, 3$ ($FG_u$ has a $1,000$-feature set, a $500$-feature set, and a $100$-feature set) from the $9,904$-feature pool independently. $500$ features are not selected from the $1,000$ selected features, and $100$ features are not selected from the $500$ selected features and the $1,000$ selected features. The 4 groups have 12 different feature sets. 12 different multi-FS methods are used to select top ranked features from the 12 different feature sets. For example, the multi-FS method sequentially uses Chi2, mutual_info_classif, f_regression, f_classif, and the RFE to select 1000 top-ranked features from the $9,904$-feature pool.

The third step of the multi-FS algorithm finally uses the RFE and the random forest to rank 1000, 500, and 100 selected features in the 4 different feature groups, respectively. Finally, 24 feature ranking sets are generated for reliable block ranking. Importantly, the 3D CNNs using 1000, 500, and 100 selected features associated with the brain images are more explainable than the traditional 3D CNN using all $32,768$ features including the irrelevant $22,864$ features associated with the black regions out of the brain images, as shown in Fig. 1.

## 5.2 Detailed Steps of Algorithm 2

12 $16 \times 16 \times 16$ feature distribution matrices are generated. 24 feature ranking matrices and 24 average feature ranking matrices are generated. The block ranking algorithm with $\check{N} = 4$, $\tilde{N} = 3$, and $\check{K} = 2$ uses the weighted average block ranking score that is defined as $\theta^{\alpha\beta\kappa}_{ijk} = w_1 c^{\alpha\beta}_{ijk} + w_2(1 - r^{\alpha\beta\kappa}_{ijk}/n) + w_3(1 - \bar{r}^{\alpha\beta\kappa}_{ijk}/Max(\bar{r}^{\alpha\beta\kappa}_{ijk}))$ where $n$ is the number of selected features, $\sum_{h=1}^{3} w_h = 1$, $0 < w_h < 1$ for $h = 1, 2, 3$, for $\alpha = 1, 2, 3$, $\beta = 1, 2, 3, 4$, $\kappa = 1, 2$, $i = 0, 1, \ldots, H - 1$, $j = 0, 1, \ldots, W - 1$, and $k = 0, 1, \ldots, D - 1$. For this simulation, $w_h = 1/3$ for $h = 1, 2, 3$.

## 5.3 Identifying Brain Areas Related to Top Blocks

The 11 different brain areas associated with the four top-ranked 3D image blocks are shown in Table 1. All 11 different brain areas in the four top-ranked blocks are associated with AD diagnosis based on cited publications, as shown in Table 5 in Appendix B. Therefore, the four top-ranked blocks have brain areas associated with AD diagnosis. For example, temporoparietal atrophy provides a useful marker of the presence of AD pathology Whitwell et al. (2011). A patient with AD characteristically reduces activity most prominently in the posterior temporoparietal and posterior cingulate cortices Womack et al. (2011). Thus, the brain area (Temporal-to-Parietal left) is associated with AD diagnosis Whitwell et al. (2011); Womack et al. (2011); Anchisi et al. (2005); Agosta et al. (2012).

Table 1: The brain areas in the top four 3D blocks for non-overlapping neighboring 3D blocks.

| Top Block | Brain Areas |
|---|---|
| (7, 11, 5) | Temporal-to-Parietal left, Frontal-to-Occipital left, hOc4la (LOC) left, hOc4lp (LOC) left, and hIP4 (IPS) left. |
| (9, 12, 4) | FG2 (FusG) left, hOc4v (LingG) left, hOc4la (LOC) left, and hOc4lp (LOC) left. |
| (5, 7, 6) | Frontal-to-Occipital left. |
| (7, 8, 11) | TE 1.1 (HESCHL) right, TE 2.2 (STG) right, TE 1.0 (HESCHL) right, and lg2 (Insula) right. |

# 6 SIMULATIONS USING OVERLAPPING NEIGHBORING 3D BLOCKS

A new 3D CNN model with two Conv3d layers (1×1×1, stride 1, padding 0) (each Conv3d layer followed by BatchNorm3d, ReLU, and a 2×2×2 max-pool) is used for a new simulation using the ADNI dataset with 982 brain images. It generates features in feature maps that are associated with overlapping neighboring 3D image blocks. Detailed steps of the new 3D image block ranking pipeline are described in Appendix C.

## 6.1 IDENTIFYING BRAIN AREAS RELATED TO TOP BLOCKS

The 11 distinct brain areas associated with the four top-ranked blocks are shown in Table 2. All 11 brain areas in the four top-ranked blocks are associated with AD diagnosis based on cited publications, as shown in Table 8 in Appendix D. For example, the atrophy of presubiculum and subiculum is the AD's earliest hippocampal anatomical marker Carlesimo et al. (2015). Predicting MCI conversion to AD combined subfield volumes and presubiculum volume are more accurate (81.1%) than the total hippocampal volume (76.7%) Khan et al. (2015). Thus, the brain area (HC-Presubiculum (Hippocampus) right) is strongly associated with AD diagnosis Carlesimo et al. (2015); Khan et al. (2015); Parker et al. (2019); Frisoni et al. (2008).

Table 2: The brain areas in the top four 3D blocks for overlapping neighboring 3D blocks.

| Top Block | Brain Areas |
|---|---|
| (4, 8, 10) | PFt (IPL) right, and 3b (PostCG) right. |
| (9, 8, 9) | HC-Presubiculum (Hippocampus) right, HC-Subiculum (Hippocampus) right, and Frontal-to-Temporal-II right. |
| (11, 6, 12) | Temporal-to-Parietal right. |
| (8, 10, 5) | CA1 (Hippocampus) left, DG (Hippocampus) left, Ph1 (PhG) left, Ph2 (PhG) left, and FG3 (FusG) left. |

## 6.2 VISUALIZING THE BRM WITH TOP-RANKED 3D IMAGE BLOCKS

Since each non-overlapping neighboring $4 \times 4 \times 4$ 3D image block contains 4 axial patches, 4 coronal patches, and 4 sagittal patches, it is necessary to help a medical doctor to easily view the 12 patches with meaningful notations to make a correct and efficient diagnosis. For instance, the "ebrains" software tool generates an axial image, a coronal image, and a sagittal image of the top-ranked $10 \times 10 \times 10$ 3D image block (9, 8, 9) located at the standard brain's world coordinates (-16.5 mm, -25.0 mm, 19.3 mm), as shown in Fig. 2. The method for generating the world coordinates of a 3D brain image block's center is described in Appendix E. A researcher may use the world coordinates to find the top-ranked block in the standard brain, then identify relevant brain areas, and finally do analysis. An enlarged axial patch, an enlarged coronal patch, and an enlarged sagittal patch displayed with relevant notations in Fig. 2 are convenient for the doctor to see three different brain areas ( (1) HC-Presubiculum (Hippocampus) right Braak & Braak (1991); Van Hoesen et al. (1991), (2) HC-Subiculum (Hippocampus) right de Toledo-Morrell et al. (2004); La Joie et al. (2012a); Yassa & Stark (2009); Burggren et al. (2008); Mueller et al. (2010b), and (3) Frontal-to-Temporal-II right Desgranges et al. (1998); Seeley et al. (2009); Liu et al. (2014); Buckner et al. (2005); Morris & Price (2001))(note: the cited publications indicate that the three brain areas are associated with AD diagnosis) in three different colors. The doctor can efficiently make a rational and interpretable diagnosis by using enlarged axial patches, enlarged coronal patches, and enlarged sagittal patches of top-ranked 3D image blocks in a patient's 3D brain image.

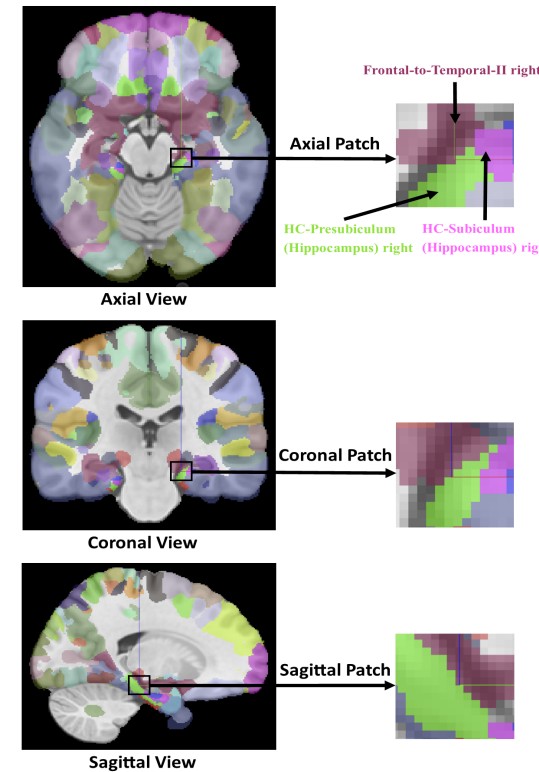

Figure 2: 3D image block (9, 8, 9) displayed in an axial image with an enlarged patch, a coronal image with an enlarged patch, and a sagittal image with an enlarged patch.

## 6.3 MODEL COMPARISON USING MULTIPLE METRICS

### 6.3.1 ACCURACIES, F1 SCORES, AUC, AND MODEL SIZES FOR AD DIAGNOSIS

A multilayer perceptron (MLP) with a 50-neuron hidden layer in a new 3D CNN model with the FF layer and a conventional 3D CNN model without the FF layer are used for the three-class 3D image classification. The 3D CNN models generate non-overlapping neighboring 3D image blocks. Tables 3 shows that all 3D CNN models with FS are better than the 3D CNN model using all 32,768 features in terms of test accuracy, F1-score, AUC, and model size.

Table 3: Performance comparison of 3D CNN models with different features for AD diagnosis.

| Features | Test Acc. | F-1 | AUC | Model Size (MB) |
|---|---|---|---|---|
| 32,768 | 0.614 | 0.574 | 0.715 | 18.77 |
| 1,000 | 0.670 | 0.647 | 0.720 | 0.67 |
| 500 | 0.655 | 0.625 | 0.743 | 0.35 |
| 100 | 0.640 | 0.597 | 0.721 | 0.09 |

### 6.3.2 ACCURACIES, F1 SCORES, AUC, AND MODEL SIZES FOR AUTISM DIAGNOSIS

286 3D brain images for autism diagnosis (binary classification) Sujana (2025) include 131 images for the autistic class and 155 images for the non-autistic class. They are resized to $64 \times 64 \times 64$ 3D images. $80\%$ of 286 3D brain images are used to generate training data, and $20\%$ of 286 3D brain images are used to generate testing data. The FS method sequentially uses f_regression, Chi2, mutual_info_classif, f_classif, and RFE to select top-ranked features. New 3D CNN models with a MLP with one hidden layer with 60 neurons are used for simulations. The new 3D CNN models generate overlapping neighboring 3D image blocks. Performance of the new 3D CNN model is evaluated under three new conditions. The first step of Algorithm 1 is not used, five FS methods

(f_regression, Chi2, mutual_info_classif, f_classif, and RFE) are used to select features, and RFE is used to rank features. Test accuracies, F1-scores and AUC values of different 3D CNN models with different features are shown in Table 4. All 3D CNN models with FS are better than the 3D CNN model using all 32,768 features in terms of test accuracy, F1-score, AUC, and model size.

Table 4: Performance comparison of 3D CNN models with different features for autism diagnosis.

| Features | Test Acc. | F-1 | AUC | Model Size (MB) |
|---|---|---|---|---|
| 32,768 | 0.690 | 0.689 | 0.705 | 18.77 |
| 3,000 | 0.776 | 0.749 | 0.741 | 1.70 |
| 1,000 | 0.793 | 0.771 | 0.762 | 0.61 |
| 500 | 0.741 | 0.710 | 0.706 | 0.32 |

## 7 CONCLUSIONS

Top-ranked features selected by the new multi-FS method are used to generate informative 3D feature distribution matrices, 3D feature ranking matrices, and 3D average feature ranking matrices. Since there is a mathematical relationship between the voxels in a $R_h \times R_w \times R_d$ 3D image block and a feature and there is a highly non-linear relationship between selected features and a final decision, the three 3D feature matrices are used by the hybrid 3D image block ranking algorithm to generate reliable 3D image block rankings. The top-ranked 3D image blocks contain brain areas associated with AD diagnosis for the two different simulations using overlapping neighboring and non-overlapping neighboring 3D image blocks. Thus, the hybrid 3D image block ranking algorithm is feasible and useful for explaining 3D image classification such as brain imaging. A top-ranked 3D image block in a BRM displayed by axial images, coronal images, and sagittal images is convenient for a medical doctor to easily understand the relationship among 3D image blocks, relevant brain areas, and the decisions of the 3D CNN with FS. Importantly, the new 3D CNN with FS is better than the traditional 3D CNN without FS in terms of test accuracy, F1-score, AUC, and model size. Thus, the new 3D CNN with the reliable FS and the hybrid 3D image block ranking is more effective than the traditional 3D CNN for explainable, accurate, and memory-efficient 3D image classification.

In summary, we have four contributions: (1) the new multi-FS algorithm is developed to effectively generate different top-ranked feature sets used for robust hybrid 3D image block ranking (the first step eliminates a large number of irrelevant features that are out of the brain regions), (2) the novel hybrid 3D image block ranking algorithm using different 3D feature matrices is created to reliably identify top-ranked 3D image blocks for effective explainable imaging, (3) the new 3D CNN using a small number of relevant top-ranked features associated with the brain regions is more explainable, more accurate and more memory-efficient than the traditional 3D CNN using all features including the irrelevant features associated with top-ranked 3D image blocks out of the brain regions, and (4) the BRM containing top-ranked 3D image blocks displayed by axial, coronal, and sagittal images showing relevant areas, such as brain areas, is useful and convenient for users, such as a doctor, to efficiently make correct and explainable decisions.

## 8 FUTURE WORKS

Since the hybrid 3D image block ranking algorithm may generate different 3D image block ranking results under different conditions, it is necessary for researchers and experts to work jointly on ranking 3D image blocks accurately based on the different BRMs, human knowledge, and publications. We will evaluate the stability of the BRM rankings across different FS method combinations. It is useful to develop explainable 3D computer vision software tools displaying top-ranked blocks by using axial, coronal, and sagittal 2D patches with informative notations, such as brain areas and their associations with a brain disease, for interpretable 3D imaging applications.

New 3D deep learning models with new reliable FS will be created. More robust hybrid 3D image block ranking algorithms using new informative matrices related to 3D image blocks and decisions, rational feature maps, effective multi-FS methods, heatmaps, and other intelligent approaches will be developed to rank 3D image blocks more accurately. Other 3D data sets, such as 3D lung cancer data sets Mader (2021), will be used for further performance analysis.

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

## A  THE 3D IMAGE BLOCK RANKING FRAMEWORK

The new 3D CNN with reliable feature selection and hybrid 3D image block ranking consists of eight components, as shown in Fig. 3.

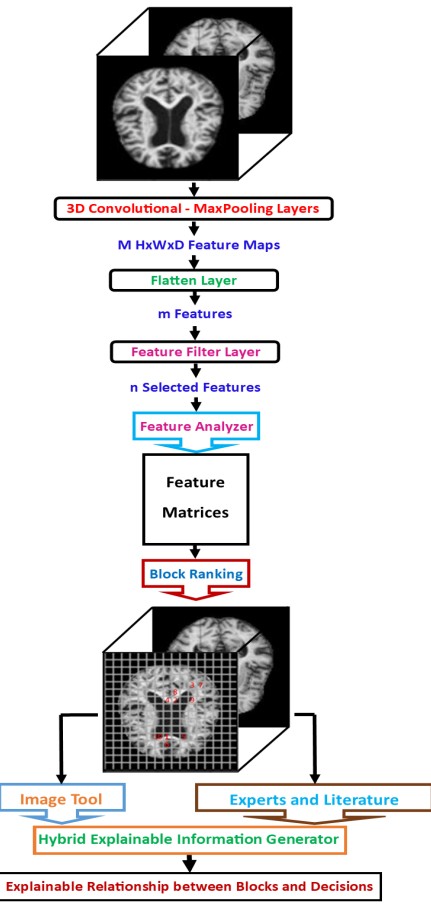

Figure 3: The 3D image block ranking framework using top-ranked feature matrices.

## B  RELATIONSHIP BETWEEN BRAIN AREAS SHOWN IN TABLE 1 AND AD DIAGNOSIS BASED ON PUBLICATIONS

All brain areas in the four top-ranked blocks are associated with AD diagnosis based on cited publications, as shown in Table 5.

## C  RANKING OVERLAPPING NEIGHBORING 3D IMAGE BLOCKS

After the first step of the hybrid FS algorithm is done, as shown in Section 5.1, the second step of the hybrid FS algorithm uses five basic FS methods (Chi2 scikit-learn developers (2025a), mutual_info_classif scikit-learn developers (2025d), f_regression scikit-learn developers (2025c), f_classif scikit-learn developers (2025b), and RFE) in 12 different ways to select 12 different feature groups $FG_u$ for $u = 0, 1, \ldots, 11$ ($FG_u$ has a $1{,}000$-feature set, a $750$-feature set, a $500$-feature set, a $250$-feature set, and a $100$-feature set) from the $9{,}904$-feature pool independently. $750$ features are not selected from the $1{,}000$ selected features. The 12 groups have 60 different feature sets. The 1st group sequentially uses Chi2, mutual_info_classif, f_regression, f_classif, and the RFE. The 2nd group sequentially uses f_classif, mutual_info_classif, f_regression, Chi2, and the RFE. The

Table 5: The brain areas shown in Table 1 associated with AD diagnosis.

| Top Block | Brain Areas |
|---|---|
| (7, 11, 5) | Temporal-to-Parietal left Babiloni et al. (2016); Rabinovici et al. (2011); Jacobs et al. (2015); Whitwell et al. (2018), Frontal-to-Occipital left Höbler et al. (2021); Josephs et al. (2014); Marcus et al. (2022); Li et al. (2021), hOc4la (LOC) left Crutch et al. (2012); Singh et al. (2015); Migliaccio et al. (2009), hOc4lp (LOC) left Maier et al. (2008); Crutch et al. (2012); Singh et al. (2015); Migliaccio et al. (2009), and hIP4 (IPS) left Crutch et al. (2012); Singh et al. (2015). |
| (9, 12, 4) | FG2 (FusG) left Rabinovici et al. (2011); La Joie et al. (2017); Mion et al. (2010), hOc4v (LingG) left Crutch et al. (2012); Ahmed et al. (2016); Nestor et al. (2003), hOc4la (LOC) left, and hOc4lp (LOC) left. |
| (5, 7, 6) | Frontal-to-Occipital left. |
| (7, 8, 11) | TE 1.1 (HESCHL) right Fitzhugh et al. (2022); Menezes et al. (2025); Tuwaig et al. (2017), TE 2.2 (STG) right Kim et al. (2020); Yeung et al. (2021); Butts et al. (2022); Menezes et al. (2025), TE 1.0 (HESCHL) right Fitzhugh et al. (2022); Menezes et al. (2025); Tuwaig et al. (2017), and lg2 (Insula) right Bonthius et al. (2005); Liu et al. (2018); Cosentino et al. (2015); Belkhiria et al. (2020). |

3rd group sequentially uses f_regression, mutual_info_classif, f_classif, Chi2, and the RFE. The 4th group sequentially uses f_classif, mutual_info_classif, and the RFE. The 5th group sequentially uses f_regression, Chi2, and the RFE. The 6th group uses the RFE. The 7th group sequentially uses f_classif, and the RFE. The 8th group sequentially uses f_regression, and the RFE. The 9th group sequentially uses Chi2, and the RFE. The 10th group sequentially uses Chi2, mutual_info_classif, and the RFE. The 11th group uses f_classif. The 12th group uses Chi2.

The third step of the hybrid FS algorithm finally uses the RFE to rank 1000, 750, 500, 250, and 100 selected features in the 12 different feature groups, respectively. Finally, 60 top-ranked feature sets are generated for reliable block ranking.

Five $16 \times 16 \times 16$ feature distribution matrices for the top-ranked 1000, 750, 500, 250, and 100 features in the 12 feature groups are generated, respectively. Five feature ranking matrices for the 1000, 750, 500, 250, and 100 features in the 12 feature groups are generated, respectively. Finally, five average feature ranking matrices for the 1000, 750, 500, 250, and 100 features in the 12 feature groups are generated, respectively.

The block ranking algorithm with $\check{N} = 12$ and $\tilde{N} = 5$ uses the weighted average block ranking score that is defined as $\theta_{ijk}^{\alpha\beta} = w_1 c_{ijk}^{\alpha\beta} + w_2(1 - r_{ijk}^{\alpha\beta}/n) + w_3(1 - \bar{r}_{ijk}^{\alpha\beta}/Max(\bar{r}_{ijk}^{\alpha\beta}))$ where $n$ is the number of selected features, $\sum_{h=1}^{3} w_h = 1$, $0 < w_h < 1$ for $h = 1, 2, 3$, for $\alpha = 1, 2, \ldots, 5$, and $\beta = 1, 2, \ldots, 12$, $i = 0, 1, \ldots, H - 1$, $j = 0, 1, \ldots, W - 1$, and $k = 0, 1, \ldots, D - 1$. For this simulation, $w_h = 1/3$ for $h = 1, 2, 3$.

8 different block feature groups $BFG_s$ for $s = 1, 2, \ldots, 8$ are generated by using the 12 different feature groups $FG_t$ for $t = 0, 1, \ldots, 11$. $BFG_1$ includes $FG_h$ for $h = 0, 1, \ldots, 11$. $BFG_2$ includes $FG_h$ for $h = 0, 1, \ldots, 10$. $BFG_3$ includes $FG_h$ for $h = 0, 1, \ldots, 9$. $BFG_4$ includes $FG_h$ for $h = 0, 1, \ldots, 8$. $BFG_5$ includes $FG_h$ for $h = 0, 1, \ldots, 7$. $BFG_6$ includes $FG_h$ for $h = 0, 1, \ldots, 6$. $BFG_7$ includes $FG_h$ for $h = 0, 1, \ldots, 4$. $BFG_8$ includes $FG_h$ for $h = 0, 1, 2$.

Table 6, as shown in Appendix C, shows 4 lists of top-ranked blocks ($List_1$, $List_2$, $List_3$, and $List_4$) of the top 10 blocks in the 4 different BRMs that are generated by using the block ranking algorithm using the 3D feature distribution matrices reliably generated by $BFG_1$, $BFG_2$, $BFG_3$, and $BFG_4$, the 3D feature ranking matrices reliably generated by $BFG_1$, $BFG_2$, $BFG_3$, and $BFG_4$, and the 3D average feature ranking matrices reliably generated by $BFG_1$, $BFG_2$, $BFG_3$, and $BFG_4$.

Table 6: Four block lists ($List_1$, $List_2$, $List_3$, and $List_4$) of the top 10 blocks with their indices.

| Rank | $List_1$ | $List_2$ | $List_3$ | $List_4$ |
|------|----------|----------|----------|----------|
| 1 | (4, 8, 10) | (4, 8, 10) | (9, 8, 9) | (4, 8, 10) |
| 2 | (5, 9, 2) | (9, 8, 9) | (4, 8, 10) | (11, 10, 7) |
| 3 | (9, 8, 9) | (11, 10, 7) | (11, 10, 7) | (9, 8, 9) |
| 4 | (8, 10, 5) | (5, 9, 2) | (5, 9, 2) | (8, 10, 5) |
| 5 | (11, 10, 7) | (11, 6, 12) | (11, 6, 12) | (11, 6, 12) |
| 6 | (6, 10, 6) | (8, 10, 5) | (8, 10, 5) | (5, 9, 2) |
| 7 | (6, 10, 5) | (6, 10, 5) | (8, 6, 5) | (7, 11, 13) |
| 8 | (11, 6, 12) | (6, 10, 6) | (6, 10, 5) | (4, 7, 11) |
| 9 | (9, 7, 5) | (10, 10, 9) | (10, 2, 5) | (8, 6, 11) |
| 10 | (9, 7, 4) | (8, 6, 11) | (4, 7, 11) | (6, 10, 5) |

In addition, Table 7, as shown in Appendix C, shows 4 other lists of top-ranked blocks ($List_5$, $List_6$, $List_7$, and $List_8$) of the top 10 blocks in the 4 different BRMs that are generated by using the block ranking algorithm using the 3D feature distribution matrices reliably generated by $BFG_5$, $BFG_6$, $BFG_7$, and $BFG_8$, the 3D feature ranking matrices reliably generated by $BFG_5$, $BFG_6$, $BFG_7$, and $BFG_8$, and the 3D average feature ranking matrices reliably generated by $BFG_5$, $BFG_6$, $BFG_7$, and $BFG_8$.

Table 7: Four block lists ($List_5$, $List_6$, $List_7$, and $List_8$) of the top 10 blocks with their indices.

| Rank | $List_5$ | $List_6$ | $List_7$ | $List_8$ |
|------|----------|----------|----------|----------|
| 1 | (4, 8, 10) | (4, 8, 10) | (9, 8, 9) | (4, 9, 12) |
| 2 | (11, 10, 7) | (9, 8, 9) | (11, 6, 12) | (10, 12, 12) |
| 3 | (8, 10, 5) | (11, 10, 7) | (9, 7, 4) | (11, 7, 13) |
| 4 | (9, 8, 9) | (11, 6, 12) | (4, 8, 10) | (4, 8, 10) |
| 5 | (8, 6, 11) | (8, 6, 11) | (10, 12, 12) | (9, 8, 9) |
| 6 | (11, 6, 12) | (8, 10, 5) | (11, 10, 7) | (9, 7, 4) |
| 7 | (11, 7, 13) | (6, 10, 5) | (4, 9, 12) | (8, 10, 5) |
| 8 | (5, 9, 2) | (7, 11, 13) | (8, 6, 11) | (8, 6, 11) |
| 9 | (6, 10, 5) | (9, 7, 5) | (10, 9, 2) | (11, 6, 12) |
| 10 | (7, 11, 13) | (5, 9, 2) | (8, 10, 5) | (10, 9, 2) |

The hybrid block ranking algorithm using equal weights ($w_l = 1/8$ for $l = 1, 2, \ldots, 8$) is used to finally generate top-ranked blocks by using the 8 different BRMs. Top 4 blocks are shown in Table 1. Block (4, 8, 10) with a $\bar{\lambda}$ of 1.88 (i.e., (1+1+2+1+1+1+4+4)/8=1.88) ranks first, and blocks (9, 8, 9), (11, 6, 12), and (8, 10, 5) rank 2nd, 3rd, and 4th, respectively. In this special case, the top 4 blocks are common blocks in all 8 block groups shown in Tables 6 and 7, as shown in Appendix C. Thus, the top 4 common blocks are important for AD diagnosis.

In summary, 8 different block feature groups with 60, 55, 50, 45, 40, 35, 25, and 15 selected feature sets are generated reliably by the hybrid FS method. Then, 8 block groups of top 10 blocks are generated by the bock ranking algorithm using the 8 different block feature groups. Finally, the 4 top-ranked common blocks are selected from the 8 block groups. Since all 11 brain areas associated with the top four common blocks are critical and useful for the 3-class AD diagnosis, the hybrid FS method and the bock ranking algorithm are feasible and useful for explaining 3D medical image classification.

## D    RELATIONSHIP BETWEEN BRAIN AREAS SHOWN IN TABLE 2 AND AD DIAGNOSIS BASED ON PUBLICATIONS

All brain areas in the four top-ranked blocks are associated with AD diagnosis based on cited publications, as shown in Table 8.

Table 8: The brain areas shown in Table 2 associated with AD diagnosis.

| Top Block | Brain Areas |
|---|---|
| (4, 8, 10) | PFt (IPL) right Greicius et al. (2004); Seeley et al. (2009); Buckner et al. (2009); Zhou et al. (2010); Sestieri et al. (2011), and 3b (PostCG) right Karas et al. (2004); Sluimer et al. (2010); Schwindt & Black (2009). |
| (9, 8, 9) | HC-Presubiculum (Hippocampus) right Braak & Braak (1991); Van Hoesen et al. (1991), HC-Subiculum (Hippocampus) right de Toledo-Morrell et al. (2004); La Joie et al. (2012a); Yassa & Stark (2009); Burggren et al. (2008); Mueller et al. (2010b), and Frontal-to-Temporal-II right Desgranges et al. (1998); Seeley et al. (2009); Liu et al. (2014); Buckner et al. (2005); Morris & Price (2001). |
| (11, 6, 12) | Temporal-to-Parietal rightJack et al. (2010); Jagust & Mormino (2011); Frisoni et al. (2010); Minoshima et al. (1997). |
| (8, 10, 5) | CA1 (Hippocampus) left Mueller et al. (2010b); Kerchner et al. (2010); Apostolova et al. (2010); La Joie et al. (2012b), DG (Hippocampus) left Small et al. (2011); Yassa et al. (2011); West et al. (2004); Iglesias et al. (2015); Mueller et al. (2010a), Ph1 (PhG) left Pennanen et al. (2004); Apostolova et al. (2006); Raj et al. (2012); Zhou et al. (2010), Ph2 (PhG) left Du et al. (2007); Zhou et al. (2010); Jagust et al. (2007); Whitwell et al. (2007), and FG3 (FusG) left Karas et al. (2004); Bokde et al. (2001). |

Firstly, PFt (IPL) right associated with Block (4, 8, 10) is important for the 3-class AD diagnosis Greicius et al. (2004); Seeley et al. (2009); Buckner et al. (2009); Zhou et al. (2010); Sestieri et al. (2011). Brain regions like the IPL are particularly vulnerable in AD, which may aid in 3-class AD diagnostic frameworks Buckner et al. (2009). The network connectivity differences between AD and other dementias show that connectivity disruptions in the IPL are prominent in AD and relevant for differential diagnosis Zhou et al. (2010). The functional specialization of the posterior parietal cortex (including the IPL) and its role in memory and attention, functions often impaired in AD Sestieri et al. (2011).

Secondly, HC-Presubiculum (Hippocampus) right associated with Block (9, 8, 9) is associated with the 3-class AD diagnosis Braak & Braak (1991); Van Hoesen et al. (1991); **?**. The presubiculum is among the early regions affected by neurofibrillary tangles and amyloid plaques Braak & Braak (1991). The pathological changes in the entorhinal cortex and adjacent regions, including the pre-subiculum, show their roles in the early stages of AD Van Hoesen et al. (1991).

Thirdly, Temporal-to-Parietal right associated with Block (11, 6, 12) is important for the 3-class AD diagnosis. Based on literature Jack et al. (2010); Jagust & Mormino (2011); Frisoni et al. (2010); Minoshima et al. (1997), Temporal-to-Parietal right is indeed associated with AD and plays a significant role in AD diagnosis, including in 3-class classification frameworks.. Dynamic biomarkers in Alzheimer's pathology include neuroimaging markers like cortical atrophy in regions including the temporal-to-parietal lobes Jack et al. (2010). MRI is used to detect cortical atrophy patterns in AD, particularly in the temporal-to-parietal regions, for clinical diagnosis Frisoni et al. (2010).

Fourthly, the CA1 (Hippocampus) left associated with Block (8, 10, 5) is critical and useful for 3-class AD diagnosis Mueller et al. (2010b); Kerchner et al. (2010); Apostolova et al. (2010); La Joie et al. (2012b). The MRI is used to evaluate hippocampal subfield atrophy to discover that the CA1 region is particularly affected in patients with MCI and AD, and CA1 atrophy is associated with memory deficits and can serve as a biomarker for early AD diagnosis Mueller et al. (2010b). The atrophy in the CA1 region occurs early in AD and is associated with amyloid deposition and hypometabolism, supporting its role as a biomarker for early AD detection La Joie et al. (2012b).

# E   THE WORLD COORDINATES (IN MM) OF A CORRESPONDING BLOCK'S CENTER IN THE STANDARD 3D BRAIN.

The center of a $4 \times 4 \times 4$ brain image block with indices $(i, j, k)$ for $i = 0, 1, \ldots, 15$, $j = 0, 1, \ldots, 15$, and $k = 0, 1, \ldots, 15$ has voxel coordinates $I^i = 4i + 2$, $J^j = 4j + 2$, and $K^k = 4k + 2$. The block contains 4 axial $4 \times 4$ patches, 4 coronal $4 \times 4$ patches, and 4 sagittal $4 \times 4$ patches.

The 2D borders of average axial brain images $\bar{S}_{axial}^{I^i}$ are $Front_{I^i}$, $Back_{I^i}$, $Left_{I^i}$, and $Right_{I^i}$. For example, $Front_{I^i} = 5$, $Back_{I^i} = 61$, $Left_{I^i} = 5$, and $Right_{I^i} = 57$ for $I^7 = 30$ for the average axial brain image in Fig. 1(d).

The 3D borders of the standard 3D brain are $Top$ = 77 mm, $Bottom$ = -78 mm, $Front$ = 78 mm, $Back$ = -114 mm, $Left$ = - 96 mm, and $Right$ = 96 mm. The 2D borders of an axial image of the standard 3D brain of the "ebrains" software tool are $front_{I^i}$, $back_{I^i}$, $left_{I^i}$, and $right_{I^i}$. Based on the standard 3D brain's axial coordinate $I$: $I = \left\lfloor (I^i \times (Bottom{-}Top)/63.0) + Top \right\rfloor$, the "ebrains" software tool is used to find $front_{I^i}$, $back_{I^i}$, $left_{I^i}$, and $right_{I^i}$. For example, $front_{I^i} = 71.5$ mm, $back_{I^i} = -104.9$ mm, $left_{I^i} = -71.5$ mm , and $right_{I^i} = 71.5$ mm for $I^7$ = 30 are found for the corresponding average axial brain image shown in Fig. 1(d).

Finally, the world coordinates (in mm) of a corresponding block's center in the standard 3D brain are $I = \left\lfloor (I^i \times (Bottom{-}Top)/63.0) + Top \right\rfloor$, $J = \left\lceil (J^j - Front_{I^i}) \times (back_{I^i} - front_{I^i})/(Back_{I^i} - Front_{I^i}) + front_{I^i} \right\rceil$, and $K = \left\lceil (K^k - Left_{I^i}) \times (right_{I^i} - left_{I^i})/(Right_{I^i} - Left_{I^i}) + left_{I^i} \right\rceil$.

