# OpenReview forum: "An Explainable 3D Convolutional Neural Network with Reliable Feature Selection and Hybrid 3D Image Block Ranking"
_ICLR.cc/2026/Conference — ICLR 2026 Conference Desk Rejected Submission_

### Official Review · Reviewer_NDVi · 2025-10-24

[review text omitted: it was posted to a different submission]

---

### Official Review · Reviewer_Bywb · 2025-10-30

**Soundness:** 1
**Presentation:** 1
**Contribution:** 2
**Rating:** 0
**Confidence:** 5

**Summary:**

The paper presents a method for creating an explainable 3D CNN by integrating feature selection (FS) with a hybrid 3D image block ranking algorithm. The core idea of linking feature importance back to specific 3D image blocks for medical diagnosis is commendable and has significant potential for clinical interpretability. The application to Alzheimer's Disease (AD) and autism diagnosis is relevant and important. However, the paper in its current form suffers from several critical weaknesses that preclude its acceptance. The most severe issues relate to the validation of the proposed method's "explainability" and the experimental setup, which raise substantial doubts about the claims made.

**Strengths:**

1. The integration of a dedicated FS layer and a multi-step hybrid algorithm for ranking 3D image blocks is a novel contribution to the field of explainable AI (XAI) for 3D imaging. The attempt to move beyond 2D heatmaps to a 3D block-based importance measure is a step in the right direction.
2. The focus on medical imaging, specifically AD and autism diagnosis, addresses a pressing need for interpretable deep learning models in healthcare. The method's output (Block Ranking Map - BRM) is designed to be clinician-friendly, which is a significant practical strength.

**Weaknesses:**

1. The paper's central claim is "explainability," but this is not validated in a meaningful way. The primary validation method is a post-hoc literature search: top-ranked blocks are identified, and then references are cited to show that the brain areas within those blocks are "associated with AD diagnosis". This is circular reasoning and does not prove that the model learned these associations.
2. Incomplete and Potentially Biased Feature Selection Process:​​ The first step of Algorithm 1  eliminates features not associated with "a rational image region defined by an expert." The rules for this elimination  appear to be manually defined based on average brain images. This process is highly subjective and risks introducing a strong confirmation bias. The model is effectively prevented from looking outside the pre-defined "rational" regions, which may ignore potentially novel or counter-intuitive biomarkers. This undermines the objective nature of the feature selection and the subsequent explainability claims.
3. The description of the methods, particularly the algorithms, is dense and difficult to follow.

**Questions:**

1. Explainability Validation:​​ Beyond correlating your results with existing literature, what concrete experiments can be conducted to provethat the top-ranked blocks are truly important for your model's decision? For example, have you considered performing an ablation study where you set the voxel values in the top-K ranked blocks to zero and measuring the change in prediction confidence?
2. ​Feature Selection Bias:​​ Could you elaborate on the process of defining the "rational image region" for the initial feature pruning? How can you ensure that this manual step does not discard features that might be genuinely informative but located in areas not traditionally associated with the disease? Would the method work without this step?
3.Algorithmic Complexity:​​ The hybrid ranking algorithm involves many parameters (number of FS methods, feature set sizes, weights w_h). Was any hyperparameter optimization performed? Can you show that the full hybrid approach is significantly better than a baseline that uses a single, strong FS method (like RFE) and ranks blocks simply by the count of selected features (a_ijkfrom matrix A)?
4.Comparison to Baselines:​​ How does the explanatory power of your BRM compare to applying a standard explanation method like Grad-CAM directly to the feature maps of your 3D CNN? A qualitative comparison of the saliency maps would be very informative.

---

> ### Author Response · Authors · 2025-11-25
>
> Q1. Many thanks for the critical question and the great suggestion! The ablation method is an effective method. In addition, we thought about few other methods.  The first human-based method was asking a large number of experienced clinicians to rank 10 top-ranked 3D blocks for AD diagnosis, and then discovering final 10 top-ranked 3D blocks by averaging their ranking scores. The second statistics-based method is randomly selecting a large number of 10-block sets at different locations, then calculating average ranking scores of the random 10-block sets, and finally comparing the average ranking scores with an average ranking score of the top-ranked blocks that generated by our Hybrid 3D Image Block Ranking Algorithm. Theoretically speaking, if different feature selection methods can work together to select truly important top-ranked features robustly and diverse feature ranking methods can work jointly to rank the top-ranked features reliably, 3D blocks associated with the feature matrices defined in Section 3.2 can be ranked effectively.
>
> Q2. The "rational image region" means a region is within a brain image. Any features associated with irrational image regions are eliminated. The remaining rational features are used for further feature selection.
>
> How can you ensure that this manual step does not discard features that might be genuinely informative but located in areas not traditionally associated with the disease?
> Answer: This manual step can ensure that candidate features are associated with regions within a brain image.
>
> Would the method work without this step?
> Answer: The method may work without this step, but many irrational features in regions such as black corner regions out of the brain image are used to build a deep learning model that is not explainable.
>
> 3. Algorithmic Complexity: The hybrid ranking algorithm involves many parameters (number of FS methods, feature set sizes, weights w_h). Was any hyperparameter optimization performed?
> Answer: Thanks for the great idea. The hyperparameter optimization was not performed. We will develop a new hyperparameter optimization in the future.
>
> Can you show that the full hybrid approach is significantly better than a baseline that uses a single, strong FS method (like RFE) and ranks blocks simply by the count of selected features (a_ijkfrom matrix A)?
> Answer: Thanks for the excellent suggestion. We will do more simulations to get reliable comparison results.
>
> 4. Comparison to Baselines: How does the explanatory power of your BRM compare to applying a standard explanation method like Grad-CAM directaly to the feature maps of your 3D CNN? A qualitative comparison of the saliency maps would be very informative.
> Answer: The standard Grad-CAM can generate heatmaps, but it is not be able to rank 3D image blocks directly. It is necessary to develop a new 3D image block ranking algorithm using heatmaps generated by Grad-CAM, and then compare our Hybrid 3D Image Block Ranking Algorithm with the Grad-CAM-based 3D image block ranking algorithm. We plan to do it in the future.

---

### Official Review · Reviewer_pVND · 2025-10-31

**Soundness:** 2
**Presentation:** 2
**Contribution:** 2
**Rating:** 2
**Confidence:** 3

**Summary:**

This paper presents an explainable 3D CNN integrating robust FS and a hybrid 3D image block ranking algorithm to address the challenge of interpreting relationships among top-ranked 3D image blocks, selected features, and model decisions. The evaluation of the model is performed on 3D brain MRI scans from the ADNI dataset.

**Strengths:**

The proposed framework involves:

* Multi-FS Procedure: eliminates irrelevant features and generates diverse feature sets.
* Informative 3D feature matrices: constructs complementary matrices to quantify block/feature importance.
* hybrid block ranking algorithm: leverages these matrices to generate a block ranking map with 3D block rankings, enabling spatial interpretation.

**Weaknesses:**

* The structure of the experiment section is difficult to follow. It does not clearly present a comparison with traditional 3D FS methods. For instance, which baseline methods were included? Revising the layout or presenting the results in a three-line table could make this section more intuitive and concise.

* While the study primarily validates the model on ADNI data for Alzheimer's disease and shows promise in assisting physicians to identify promising candidates, further experimentation on more diverse datasets is needed to enhance its generalizability.

**Questions:**

* the traditional 3D CNN without FS is used as a baseline, but comparisons to state-of-the-art explainable 3D models (Grad-CAM for 3D, attention-based methods) are missing, It is unclear if the proposed approach outperforms existing interpretable alternatives.

* cPlease clarify the logical progression of the innovations in Section 3.2. Specifically, explain how they collectively form a cohesive, step-by-step framework that culminates in the Hybrid 3D Image Block Ranking Algorithm presented in Section 4.

---

> ### Author Response · Authors · 2025-11-25
>
> Q1: Thanks for your excellent suggestion! The explainable 3D models such as Grad-CAM for 3D don’t rank 3D image blocks; we cannot compare our block ranking method directly with them. In the future, we will implement a new 3D image block ranking method using a heatmap such as Grad-CAM based heatmaps, then compare our block ranking method with the new method.
>
> Q2:	During training, voxels in a 3D image block with the receptive fields (Rh, Rw, Rd) of an input 3D image are used to generate K features in K final feature maps. After an offline feature selection, k features associated with the 3D image block are selected for 0<=k<=K. A  3D image block with k1 selected features is more important for making a decision than another 3D image block with k2 selected features for three conditions including (1) k1>k2, (2) a top ranked feature among k1 selected features ranks higher than a top ranked feature among k2 selected features, and (3) an average ranking of k1 selected features is higher than an average ranking of k2 selected features. Therefore, 3D feature accumulation matrix and 3D feature distribution matrix associated with the first condition, 3D feature ranking matrix associated with the second condition, and 3D average feature ranking matrix associated with the third condition are newly defined in Section 3.2. Finally, the Hybrid 3D Image Block Ranking Algorithm presented in Section 4 uses the three factors related to the three conditions to rank 3D image blocks reliably by using different selected feature sets generated by the multi-FS algorithm.

---

### Official Review · Reviewer_oiGM · 2025-11-01

**Soundness:** 3
**Presentation:** 4
**Contribution:** 2
**Rating:** 6
**Confidence:** 3

**Summary:**

This paper proposes an explainable framework for 3D CNNs, combining a robust feature selection (FS) method and a new hybrid 3D image block ranking algorithm. It first uses a multi-step FS algorithm (Algorithm 1) to remove "irrelevant features" (e.g., outside the brain) and generate feature subsets. Then three 3D feature matrices (distribution, ranking, average-ranking) quantify feature importance, and a hybrid algorithm (Algorithm 2) uses them to create a "Block Ranking Map" (BRM) for spatial explainability. Key contributions: (1) Multi-FS and hybrid algorithms. (2) BRM correlates with AD-relevant anatomy. (3) 3D CNN with FS outperforms traditional ones in accuracy, F1-score, AUC and memory. (4) Validation on ADNI (3-class AD) and autism datasets.

**Strengths:**

The paper is well-structured, clearly outlining the problem (3D CNN explainability), the proposed solution (FS Layer + Hybrid Ranking), technical details (Sections 3, 4), and experimental validation (Sections 5, 6).

**Weaknesses:**

$$Technical:$$

1. The paper repeatedly refers to a new "FS Layer". However, based on the descriptions in Algorithm 1, 2, and Section 5.1, the feature selection (FS) appears to be a post-hoc process performed offline on features extracted from a pre-trained 3D CNN. This process (esp. Alg 1, Step 1) uses the entire training dataset and expert knowledge (average brain images ). This is not a "layer" in the sense of an end-to-end trainable component. This ambiguity confuses post-hoc explanation with an intrinsically explainable model.

2. Step 1 of Algorithm 1, which is critical to the method's success (removing ~70% of features ), relies on an "expert-defined" region created by averaging the training set images. This raises two concerns: (a) Does this hard feature elimination risk overfitting to the training data's specific morphology? (b) For 3D images without a clear "object of interest" (e.g., fluid dynamics), this first step would fail, limiting the method's generality.

$$Experimental:$$

3. The paper compares the "3D CNN with FS" to a "3D CNN without FS". However, the "without FS" baseline is a very simple 3D CNN (2 Conv3d layers) followed by an MLP that must handle 32,768 features. This architecture is highly susceptible to overfitting, and its poor performance (e.g., 0.614 Test Acc ) is unsurprising. A fairer, stronger baseline would apply strong regularization (e.g., Dropout) to the MLP when processing all 32k features.

4. As an XAI paper, it is a major omission to not compare the proposed BRM against any existing 3D XAI method (e.g., 3D Grad-CAM, 3D Guided Backpropagation, 3D LIME/SHAP). The paper claims its method is superior but provides no experimental evidence to support this.

5. Evaluation is limited to ADNI and one autism dataset. Testing on other 3D medical domains (e.g., lung CT, cardiac MRI) would better demonstrate generalizability.

$$Writing:$$

6. The core of Alg 2, the "block ranking score" θ, is defined as a weighted average. In both simulations, the weights are simply set to wh= 1/3. How are these weights chosen? Are they fixed? Tuned? The choice of weights is critical to the final ranking, and this is not discussed.

7. The paper mentions resizing ADNI images from various sizes (e.g., 192* 192 *160) down to 64 * 64*64. This is an aggressive downsampling that will severely distort anatomical structures and lose key information, which likely contributes to the very low baseline accuracy. This decision needs to be justified in detail

8. The “ebrains” tool is mentioned but not described in terms of accessibility or reproducibility.

**Questions:**

$$Technical:$$

1. The authors must clarify the exact implementation of the "Feature Selection Layer". Is this a layer that operates during training (e.g., a learnable gating layer)? If so, please provide details on its forward and backward propagation. If it is a post-hoc process applied after feature extraction, please revise the terminology throughout the paper (e.g., "FS process" or "FS module") to avoid confusion.

2. How stable is the BRM ranking across different FS method combinations? Could you report Jaccard similarity or Kendall’s τ between top-k block lists from different FS runs?

$$Experimental:$$

3. To prove the superiority of the proposed "CNN + FS" model, the authors need to compare it against a stronger baseline. Please compare your model (e.g., MLP on 1000 features ) against an MLP baseline trained on all 32,768 features but which includes strong, optimized regularization (e.g., tuned Dropout rate and L2 penalty).

$$Writing:$$

4. Consider releasing BRM visualization code or pseudocode in the appendix.

---

> ### Author Response · Authors · 2025-11-25
>
> Q1. Thanks for this great suggestion. The "FS Layer" doesn’t operate during training. The final Conv3d layer extracts 8 16x16x16 feature maps that are converted to 32,768 flatten features. 785 new training data with the 32,768 features are used for offline FS. Our newly revised paper changes "FS Layer" into ``Feature Filter (FF) Layer" (Fig. 3 also uses Feature Filter Layer), and clearly states “(3) the FF layer for generating $n$ top-ranked features from the $m$ flattened features by using top-ranked feature index numbers (note: the top-ranked feature index numbers have been identified by an offline FS method)” on Page 4.
>
> Q2. We observed that different multi-FS algorithms produced different top-ranked feature sets, and consequently yielded different BRM ranking scores. In this study, we did not explicitly evaluate the stability of the BRM rankings across different FS method combinations. To address this limitation, we will compute the Jaccard similarity between the top-ranked feature sets obtained from different multi-FS methods. Our newly revised paper adds “We will evaluate the stability of the BRM rankings across different FS method combinations.” in Future Works on Page 9.
>
> Q3. Thanks for the good suggestion! We will do more simulations under different conditions to get robust comparison results against a stronger baseline.
>
> Q4. Our paper showed manually visualized BRM. We will implement a new BRM visualization code, and add it in the appendix.

---

### Note · Program_Chairs · 2026-01-17
**Submission Desk Rejected by Program Chairs**

The following references in this submission do not refer to real documents and/or have major errors in bibliographic information:

 S. Ahmed, M. Irish, C. Loane, I. Baker, M. Husain, S. Thompson, L. Clare, J. R. Hodges, and O. Piguet. Neuroimaging correlates of posterior cortical atrophy: A case series and review. Cortex, 85:26-37, 2016. doi: 10.1016/j.cortex.2016.09.019.
J. L. Whitwell, J. Graff-Radford, T. D. Singh, D. A. Drubach, M. M. Machulda, D. W. Dickson, and K. A. Josephs. Imaging correlates of alzheimer's disease and other neurodegenerative disorders: A review. NeuroImage: Clinical, 20:1178-1191, 2018. doi: 10.1016/j.nicl.2018.10.020.